# Determinants of breastfeeding self-efficacy among postpartum women in rural China: A cross-sectional study

Linhua Li[1☉], Yuju Wu[1☉], Qingzhi Wang[1], Yefan Du[1], Dimitris Friesen[2], Yian Guo[2], Sarah-Eve Dill[2], Alexis Medina[2], Scott Rozelle[2], Huan Zhou[1]*

1 Department of Health Behavior and Social Medicine, West China School of Public Health and West China Fourth Hospital, Sichuan University, Chengdu, Sichuan, China, 2 Stanford Center on China's Economy and Institutions, Stanford University, Stanford, California, United States of America

☉ These authors contributed equally to this work.
* zhouhuan@scu.edu.cn

**Data Availability Statement:** All relevant data are within the manuscript and its Supporting information files.

## Abstract

### Background

Breastfeeding self-efficacy is known to positively influence breastfeeding behaviors. While previous research has studied the determinants of breastfeeding self-efficacy in general, these determinants are unstudied among postpartum women in rural China. This study aims to describe the breastfeeding self-efficacy of postpartum women in rural China and identify determinants of breastfeeding self-efficacy using the Dennis breastfeeding self-efficacy framework.

### Methods

Using a multi-stage random cluster sampling design, cross-sectional survey data were collected from 787 women within the 0–6 months postpartum period in 80 rural townships. Surveys collected data on breastfeeding self-efficacy, characteristics related to the Dennis breastfeeding self-efficacy framework, and demographic characteristics. Multiple linear regression analysis was used to identify determinants of breastfeeding self-efficacy.

### Results

Participants reported a moderate level of breastfeeding self-efficacy, with an item mean score of 3.50. Self-efficacy was lowest for exclusive breastfeeding. Breastfeeding attitudes ($\beta = 0.088$, $P < 0.001$), breastfeeding family support ($\beta = 0.168$, $P < 0.001$), and social support from significant others ($\beta = 0.219$, $P < 0.001$) were positively associated with breastfeeding self-efficacy. Breastfeeding problems, including trouble with latching ($\beta = -0.170$, $P < 0.001$), not producing enough milk ($\beta = -0.148$, $P < 0.001$), and milk taking too long to secrete ($\beta = -0.173$, $P < 0.001$) were negatively associated with breastfeeding self-efficacy.

### Conclusion

The findings indicate that positive attitudes, breastfeeding family support and social support contribute to greater breastfeeding self-efficacy in rural China, whereas difficulties with

**Funding:** This work was supported by the National Natural Science Foundation of China [grant no. 71874114] and Health Commission of Sichuan Province [grant no. 19PJ072] in the form of grants to HZ. The funders had no role in study design, data collection and analysis, decision to publish, or preparation of the manuscript.

**Competing interests:** The authors have declared that no competing interests exist.

breastfeeding are associated with reduced self-efficacy. Researchers and practitioners should investigate effective strategies to improve social support and family support for breastfeeding, promote positive attitudes towards breastfeeding, and provide women with actionable solutions to breastfeeding problems.

## Introduction

Despite the established benefits of breastfeeding, current infant breastfeeding rates remain suboptimal in low- and middle-income countries (LMICs). Breastfeeding not only provides optimal infant nutrition but also has short- and long-term health benefits for infants and mothers [1], prompting the World Health Organization (WHO) and the United Nations International Children's Emergency Fund (UNICEF) to strongly recommend that mothers initiate breastfeeding within one hour of birth, exclusively breastfeed their infants for the first six months of life and maintain breastfeeding for the first two years of life [2]. Despite these recommendations, a recent study has found that only 37% of infants under six months in LMICs were exclusively breastfed [3], well below the WHO 90% benchmark [4]. To effectively address the suboptimal breastfeeding situation, it is necessary to identify the key modifiable factors that influence breastfeeding behavior.

The international literature has shown that breastfeeding self-efficacy (BSE) is one of the most crucial, modifiable factors influencing postpartum women's breastfeeding behavior [5, 6]. BSE is derived from the self-efficacy concept of Bandura [7]. Dennis developed a framework for BSE in 1999 [8], defining BSE as a mother's perceived ability to breastfeed her child. In Dennis's framework, BSE influences a mother's breastfeeding decisions, including the decision to breastfeed, how much effort should be given to breastfeeding, and how to respond to challenges during breastfeeding [5, 6, 8]. High BSE has also been associated with greater likelihood of exclusive breastfeeding in the first 6 months after birth [9].

Given the established links between BSE and breastfeeding outcomes, it is important to understand what factors may contribute to BSE. In Bandura's self-efficacy framework and Dennis's BSE framework, four factors are posited to modify self-efficacy: a.) Performance accomplishments; b.) Vicarious experiences; c.) Verbal persuasion; and d.) Emotional arousal [7, 8] (Fig 1). Performance accomplishments refer to the expectation that one's future outcomes will be similar to past experiences. Thus, successful breastfeeding (positive performance accomplishments) may increase BSE, whereas repeated failures or problems (negative performance accomplishments) may diminish BSE. Vicarious experiences, which refer to seeing others succeed or fail in a breastfeeding, can create beliefs about one's own skills and abilities, thus

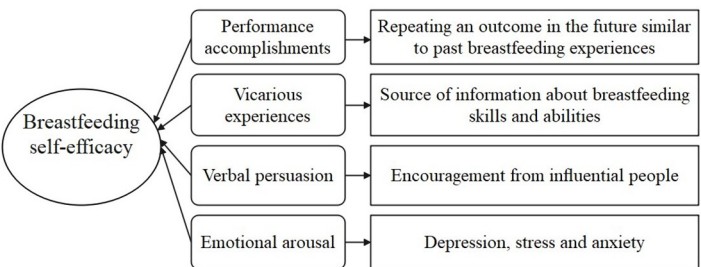

**Fig 1. Diagrammatic representation of the Dennis breastfeeding self-efficacy framework.**

impacting BSE. Verbal persuasion, or encouragement from influential people such as friends and family, can also promote the BSE of mothers. Finally, emotional arousal (such as depression, stress, or anxiety) may influence a mother's self-efficacy in general and BSE in particular [8]. Empirical studies have found evidence linking factors such as positive breastfeeding experiences, breastfeeding knowledge, breastfeeding attitudes, social support, and postpartum depression, to BSE [10, 11]. However, negative performance accomplishments (i.e., difficulties in successfully breastfeeding) have been less studied internationally, leaving a gap to be filled in the literature.

In addition, because cultural context may influence self-efficacy and related factors, there is a need for more studies of BSE in various LMIC settings, particularly those with documented low rates of breastfeeding. One understudied setting with low rates of breastfeeding is rural China. China is the most populous country in the world, and more than 60% of the country's population lives in rural areas. Although the weighted prevalence for breastfeeding in China is 79.6%, only 20.8% of infants are breastfed exclusively for six months [12]. In rural China, exclusive breastfeeding rates among children younger than six months are even lower: a survey in 26 poor, rural counties of China found that the rate of exclusive breastfeeding was only 58.3% among newborn infants, declining further to 29.1% in those aged three to four months and 13.6% in those aged five to six months [13].

Little is known about BSE in rural China; however, previous studies have measured the level of BSE among postpartum women in urban areas of China. These studies have found that the mean scores of items on the Breastfeeding Self-Efficacy Scale in Hong Kong, Shanghai and Guangzhou were 3.92 [14], 3.67 [15] and 3.38, [16] respectively. Interventions targeting BSE have also been shown to be effective in promoting breastfeeding practices among postpartum women in urban areas in China [17, 18]. However, because self-efficacy is a psychological category that is related to many socio-cultural factors, the results of international studies and studies in China's urban areas are not necessarily representative of rural China, and the lack of studies examining BSE among postpartum women in rural China presents another gap in the existing literature. Given the low rates of breastfeeding in rural China and the important role that BSE plays in breastfeeding behavior in the international literature, research on BSE in rural China is needed to inform public health policies and improve breastfeeding outcomes. Therefore, this study aims to describe the BSE of postpartum women in rural China, and to identify the determinants of BSE among postpartum women in rural China based on the Dennis BSE framework.

## Methods

### Design

This study was conducted using a multi-stage random cluster sampling design and cross-sectional survey to assess BSE among women 0–6 months postpartum in rural China and to identify determinants of BSE based on the Dennis BSE framework. This research design was based on two main considerations. First, a large-scale survey gives the research team adequate statistical power to identify population-level trends and correlations. Second, the survey research method can help us to obtain first-hand research data directly from postpartum women in rural areas of China.

### Setting

This study was conducted in poor rural areas of one prefecture in Sichuan Province, China. Located in the interior of southwest China, Sichuan ranks 18th of the 31 provinces in terms of Gross Domestic Product (GDP) per capita [19]. Almost half (46.21%) of Sichuan's population

is rural, with an average rural disposable income of USD 2,461, lower than the national average rural disposable income of USD 2,647 [19]. The sample prefecture was selected because it is relatively representative of Sichuan's rural population: 52% of the prefecture's population are rural residents, close to that of Sichuan overall (46.21%) [20]. Within the prefecture, there is one general hospital and one maternal and child health hospital in each county, a government-funded hospital in each township, and a doctor trained in medicine and public health in each village.

## Sample

The research team sampled rural mothers within the 0–6 months postpartum period following a multi-stage cluster sampling protocol. First, four nationally-designated poverty counties were selected within the sample prefecture. Second, sample townships were chosen within each sample county. To select townships representative of typical rural areas, the sampling frame excluded non-rural townships and rural townships with populations of less than 10,000. Of the remaining townships, 20 townships per county were randomly selected by a computer-generated random numbers method, resulting in a total of 80 townships. Finally, a list of all mothers with registered births within the 6 months prior to the survey was obtained from the township health center in each sample township. A total of 842 postpartum women were identified and invited to participate in the study by the research team, with the assistance of local township and village doctors. In total, 55 eligible postpartum women failed to enroll in the study due to out-migration or travel at the time of the survey, intellectual disability or mental illness that prevented ability to give informed consent, or refusal to participate. Of the 842 lactating postpartum women who enrolled in this study, 787 postpartum women completed all aspects of the questionnaire, with a response rate of 93.5%.

## Measurements

**Outcome measurement.** The "BSE" outcome was measured using the Chinese (Mandarin) version of the Breastfeeding Self-Efficacy Scale-Short Form (BSES-SF), which has been validated in mainland China with a Cronbach's alpha of 0.94 [5]. The BSES-SF is comprised of 14 positively-worded statements regarding mothers' self-efficacy in their ability to breastfeed [16, 21]. In addition to these 14 items, we also included two items ("I can always exclusively breastfeed without my child receiving even a drop of water;" and "I can always stop someone from trying to feed my child liquids or foods other than breast milk before six months of age") adapted from a BSE scale by Boateng et al. [22] to better measure maternal self-efficacy in exclusive breastfeeding in rural China. Two experienced independent researchers fluent in English and Chinese translated the two additional items into Chinese (Mandarin) before addition. Combining the 14 items from the BSES-SF with the two items from Boateng et al [22], the BSE questionnaire in this study includes 16 items measured on a 5-point Likert-type scale with answers ranging from not at all confident (1) to always confident (5). Responses were summed to calculate a total score ranging from 16 to 80, with higher scores indicating higher BSE. In the present study, the Cronbach's alpha for this questionnaire is 0.88.

**Covariate measurements.** After an extensive literature review, the research team developed a "Breastfeeding Problems Questionnaire" that encompasses the most prevalent problems associated with breastfeeding [23–31]. This questionnaire contains 19 items that measures concerns or difficulties of mothers during the first two weeks of breastfeeding. Each question in the questionnaire was answered with a "yes" or "no" answer. All items of the Breastfeeding Problems Questionnaire are presented in S1 Table.

Breastfeeding attitudes were assessed using the Iowa Infant Feeding Attitude Scale (IIFAS) [32]. This 17-item scale covers various dimensions of infant feeding attitudes, which mothers were asked to rank on a 5-point Likert scale from strongly disagree (1) to strongly agree (5). Higher scores indicate a more positive attitude to breastfeeding. The tool has been found to be reliable and valid, with a Cronbach's alpha of 0.62 in mainland China [33]. In the present study, the Cronbach's alpha is 0.56.

The "Breastfeeding Knowledge Questionnaire" was adapted and modified from the Breast-feeding Knowledge Questionnaire-Short Form (BFKQ-SF) [34] by the research team to fit the setting of rural China. This questionnaire has 12 items. Each correct answer is scored as 1, while wrong or unclear answers are scored as 0. The total score ranges from 0–12, with higher scores indicating greater knowledge about breastfeeding. All items of the Breastfeeding Knowl-edge Questionnaire can be found in S2 Table.

Family support for breastfeeding perceived by the mother was measured using a scale designed by Zhu et al. [35]. The scale contains nine items, with response ranked on a Likert scale ranging from strongly disagree (1) to strongly agree (4). Mothers with scores of 27 or higher are considered to have positive support. The scale has been evaluated for reliability and validity and has been proven to be effective at measuring family support for the breastfeeding of mothers [35]. In the present study, the Cronbach's alpha is 0.78.

The Multidimensional Scale of Perceived Social Support (MSPSS) was used to measure per-ceived social support (unrelated to breastfeeding) from family, friends and significant others [36]. This scale contains 12 items, with responses ranked on a 7-point Likert scale from strongly disagree (1) to strongly agree (7). Scores range from 12 to 84, with higher scores indi-cating higher levels of perceived social support [37]. In the present study, the Cronbach's alphas for the MSPSS total scale and family, friends, and significant others subscales are 0.89, 0.82, 0.85, and 0.80, respectively.

The 21-item version of the Depression Anxiety Stress Scales (DASS-21) is a 21-item ques-tionnaire first presented by Lovibond in 1995 that uses seven questions to measure each of the symptoms of stress, anxiety, and depression [38]. This questionnaire is designed as a Likert questionnaire, with item scores ranging from zero to three indicating different levels of sever-ity of a particular symptom experienced over the past week. In the present study, the Cron-bach's alpha for the DASS-21 total scale and depression, anxiety, and stress subscales are 0.91, 0.82, 0.71, and 0.81, respectively.

The Edinburgh Postnatal Depression Scale (EPDS) is a 10-item instrument developed to identify mothers who may be experiencing postpartum depression [39]. Each item has four possible answers, with item scores ranging from zero to three. Total possible scores range from 0 to 30, with higher scores indicating a more elevated risk for postpartum depression [39]. The cutoff point for assessing depression varies by country, with an appropriate EPDS cutoff score of >10 for postnatal depression in China [40]. In the present study, the Cronbach's alpha is 0.79.

**Participant characteristics.** Participant characteristics were collected through a demo-graphic questionnaire developed by the research team. Characteristics included mother's age, parity (primipara or multipara), marital status, education level, occupation, household eco-nomic level, mode of delivery (vaginal birth or cesarean section), and infant age in months. To assess the economic level of the household, we created a household asset index using polycho-ric principal component analysis based on whether the household owned the following assets: tap water, water heater, washing machine, computer, broadband, refrigerator, air conditioner, motorcycle, car, heating facility, toilet facility, and cooking fuel.

**Data collection.** Data were collected through a large-scale cross-sectional survey. To ensure the accuracy and consistency of our data collection, a uniform training session was

provided to enumerators; in addition, following Li et al. [15], a pilot study was conducted among twenty participants in two non-sample townships to ensure the survey was appropriate and understandable for rural mothers in the study area. We used Survey Solutions Version 21.01 (The World Bank Group, Washington, DC) to administer the survey.

All data obtained were verified three times before being officially recorded and used for analysis. First, after the initial survey, each enumerator's data were checked by a separate enumerator before leaving the township to ensure its integrity and accuracy. The data were then verified a second time by a member of the research team to confirm that the questionnaire was filled out completely and without errors. Finally, the data were handed over to another member of the research team that managed the online survey data for a third verification to confirm that there were no errors. If any errors were found, the corresponding questionnaire was rejected, and the enumerator interviewed the mother again to answer the survey questions.

## Data analysis

All data analyses were performed using Stata 15.1 (StataCorp, College Station, TX). Descriptive statistics were used to analyze BSE; breastfeeding problems; breastfeeding knowledge; breastfeeding attitude; breastfeeding family support; social support; postpartum depression, anxiety, and stress symptoms; and demographic characteristics. Kolmogorov-Smirnov tests were conducted for all continuous variables to assess the distribution of the data. To identify potentially significant influencing factors, different analyses were applied according to the characteristics of the independent variables: a one-sided independent sample t-test was employed to compare BSE between two groups, and analysis of variance was used to compare BSE among three or more groups. Spearman's correlation coefficients were conducted to test the correlation between BSE and continuous variables which were not normally distributed. Multiple linear stepwise regression models were used to perform multivariate analysis and identify the determinants of BSE. *P* values below 0.05 were considered statistically significant.

## Ethical considerations

This study received ethical approval from the Stanford University Institutional Review Board (Protocol 44312) on October 28, 2019 and the Sichuan University Ethical Review Board (Protocol K2019029) on July 15, 2019. All participants provided written, informed consent to participate in the study before the survey began. Participants were given guarantees of voluntary participation and confidentiality.

## Results

### Characteristics of the participants

Demographic characteristics of the 787 postpartum women who participated in the study are shown in Table 1. The results of the Kolmogorov-Smirnov for maternal age suggest that the distribution is not normal ($Z = 2.314$, $P < 0.05$). The median age of the postpartum women was 27 years ($IQR = 24 \sim 31$). The majority of the participants were multipara (69.5%). Almost all participants were married (98.7%), and 48.8% of mothers had graduated from junior high school. Only 20.1% of mothers were working or self-employed, and 26.2% of mothers had a very low household economic level. In addition, 55.3% of the mothers had given birth by cesarean section. About half (45.9%) of babies were aged 1–3 months, while 18.2% were under 1 month and 36.0% were 4–6 months.

**Table 1. Descriptive characteristics of postpartum women in rural China (N = 787).**

| Domain | Characteristics | N (%) |
|---|---|---|
| Socio-demographic | Age (years)[a] | |
| | 18–30 | 583 (74.1) |
| | ≥31 | 204 (25.9) |
| | Marital status | |
| | Married/Partner | 777 (98.7) |
| | Single | 10 (1.3) |
| | Education | |
| | Lower than junior high school | 90 (11.4) |
| | Junior high school | 384 (48.8) |
| | Senior high school | 134 (17.0) |
| | College/university or higher | 179 (22.7) |
| | Occupation | |
| | Farming | 6 (0.8) |
| | Working/self-employed | 153 (20.1) |
| | Not working | 623 (79.2) |
| | Household economic level [b] | |
| | Very low | 206 (26.2) |
| | Low | 192 (24.4) |
| | Moderate | 197 (25.0) |
| | High | 192 (24.4) |
| Birth-related | Parity | |
| | Primipara | 240 (30.5) |
| | Multipara | 547 (69.5) |
| | Mode of delivery | |
| | Vaginal delivery | 352 (44.7) |
| | Cesarean section delivery | 435 (55.3) |
| | Infant age (months) | |
| | 0–1 | 143 (18.2) |
| | 1–3 | 361 (45.9) |
| | 4–6 | 283 (36.0) |

Notes.

[a] We divided mothers into two age groups using 30 years as a node, following the methods of a previous study of BSE in urban China by Zhu et al. [41].

[b] Household economic level was operationalized based on participants' familial possession of twelve different household assets using principal component analysis, including tap water, water heater, washing machine, computer, broadband, refrigerator, air conditioner, motorcycle, car, heating facility, toilet facility, and cooking fuel, which was then divided into four different levels using quartiles.

## Description of BSE of the postpartum woman in rural China

The BSE scores of the study participants are presented in Table 2. The results of the Kolmogorov-Smirnov test suggest that the distribution of BSE scores is normal ($Z = 1.082$, $P > 0.05$). At the time of the survey, the average BSE score among the participants was 55.95 ($SD = 8.92$), and the mean score for each item was 3.50 ($SD = 0.56$). Mothers were most confident with "dealing with the fact that breastfeeding can be time-consuming," and were least confident with "being able to exclusively breastfeed without their child receiving even a drop of water." The data show that respondents were relatively less confident in their breastfeeding technique

**Table 2. Participant responses to the Breastfeeding Self-Efficacy scale (N = 787).**

| Item | Mean (SD) |
|---|---|
| **Interpersonal Concerns** | |
| I can always deal with the fact that breastfeeding can be time-consuming. | 4.05 (0.63) |
| I can always keep wanting to breastfeed. | 3.96 (0.74) |
| I can always comfortably breastfeed with my family members present. | 3.57 (0.94) |
| **Breastfeeding Technique** | |
| I can always ensure that my child is properly latched on for the whole feeding. | 3.86 (0.79) |
| I can always tell when my child is finished breastfeeding. | 3.68 (0.87) |
| I can always successfully cope with breastfeeding like I have with other challenging tasks. | 3.62 (0.84) |
| I can always be satisfied with my breastfeeding experience. | 3.56 (0.86) |
| I can always continue to breastfeed my child for every feeding. | 3.46 (0.93) |
| I can always manage the breastfeeding situation to my satisfaction. | 3.44 (0.89) |
| I can always manage to keep up with my child's breastfeeding demands. | 3.42 (1.02) |
| I can always determine that my child is getting enough milk. | 3.37 (0.98) |
| I can always manage to breastfeed even if my child is crying. | 3.23 (0.97) |
| I can always breastfeed my child without using formula as a supplement. | 3.20 (1.08) |
| I can always finish feeding my child on one breast before switching to the other breast. | 3.10 (1.06) |
| I can always stop someone from trying to feed my child liquids or foods other than breast milk (e.g. infant formula, milk, porridge, juice, tea [whatever is given]), before 6 months of age. | 3.61 (1.04) |
| I can always exclusively breastfeed without my child receiving even a drop of water. | 2.84 (1.04) |
| **Mean score of each item** | 3.50 (0.56) |
| **Total score** | 55.95 (8.92) |

Note. The Breastfeeding Self-Efficacy Scale includes the Interpersonal Thoughts subscale and the Technique subscale, with each item rated on a 5-point Likert-type scale (1 = not at all confident to 5 = always confident).

(mean = 3.41) and relatively more confident in dealing with interpersonal concerns in breastfeeding (mean = 3.86). Overall, postpartum women in rural China reported a moderate level of BSE.

The relationships between demographic characteristics and BSE are shown in Table 3. The results find that none of the demographic variables were significantly associated with BSE.

## Explanatory variables related to Dennis's BSE framework

**Breastfeeding problems.** Table 4 presents the descriptive statistics for breastfeeding problems and their univariate correlations with BSE. We find that in the first two weeks of breastfeeding, 44.1% of the postpartum women felt back pain; 28.5% reported problems with their child sucking or latching on properly; and 47.1% perceived insufficient milk supply. Among 17.2% of mothers, the side effects of cesarean sections affected breastfeeding; 16.1% reported that their child was distracted or disinterested in breastfeeding; and 37.2% had problems with slow milk secretion. Only 15.0% of mothers reported that their child was not growing fast enough or losing too much weight, and 17.0% had problems with clogged milk ducts.

The univariate analysis shows that eight of these breastfeeding problems had a significant association with BSE (Table 4). Specifically, these problems included back pain ($P = 0.012$), child latching poorly ($P < 0.001$), insufficient breast milk ($P < 0.001$), cesarean delivery affecting breastfeeding ($P < 0.001$), baby not interested or distracted by breastfeeding ($P < 0.001$), secreting breast milk too slowly ($P < 0.001$), baby not growing fast enough or losing too much weight ($P = 0.025$), and clogged milk duct ($P = 0.007$).

**Table 3. Differences in breastfeeding self-efficacy among various demographic sub-groups (N = 787).**

| Domain | Characteristics | Breastfeeding self-efficacy | | |
| --- | --- | --- | --- | --- |
| | | Mean (SD) | F | P-Value |
| Socio-demographic | Age (years) | | 0.85 | 0.356 |
| | 18–30 | 55.77 (8.77) | | |
| | ≥31 | 56.44 (9.32) | | |
| | Marital status | | 0.90 | 0.344 |
| | Married/Partner | 55.92 (8.87) | | |
| | Single | 58.60 (12.27) | | |
| | Mother's education | | 1.84 | 0.139 |
| | Lower than junior high school | 57.92 (8.56) | | |
| | Junior high school | 55.52 (8.97) | | |
| | Senior high school | 55.63 (8.55) | | |
| | College/university or higher | 56.09 (9.18) | | |
| | Mother's occupation | | 0.54 | 0.583 |
| | Farming | 54.50 (8.34) | | |
| | Working/self-employed | 55.35 (8.06) | | |
| | Not working | 56.11 (9.13) | | |
| | Family economic level | | 2.10 | 0.098 |
| | Very low | 56.48 (8.30) | | |
| | Low | 56.65 (9.03) | | |
| | Moderate | 54.62 (8.60) | | |
| | High | 56.02 (9.64) | | |
| Birth-related | Mother's parity | | 0.77 | 0.381 |
| | Primipara | 55.53 (8.98) | | |
| | Multipara | 56.13 (8.89) | | |
| | Mode of delivery | | 1.24 | 0.267 |
| | Vaginal delivery | 56.34 (8.81) | | |
| | Cesarean section delivery | 55.63 (9.00) | | |
| | Infant age (months) | | 0.41 | 0.663 |
| | 0–1 | 56.55 (8.79) | | |
| | 1–3 | 55.76 (8.92) | | |
| | 4–6 | 55.88 (9.00) | | |

Note. F = value of ANOVA.

**Other covariates.** Table 5 presents the descriptive statistics of the Dennis BSE Framework variables excluding breastfeeding problems. The results of the Kolmogorov-Smirnov test suggest that the distribution of all these variables is not normal ($P < 0.05$); we therefore use the median and interquartile range for our subsequent analysis. At the time of the survey, the median score for breastfeeding knowledge was 7, indicating a "good" level of knowledge. For the IIFAS, the median score was 62, indicating that breastfeeding attitudes in rural areas of China were at a medium to high level, with a high proportion of women holding positive breastfeeding attitudes. Breastfeeding family support had a median score of 34, implying that most women had positive family support for breastfeeding. The social support score averaged 66 indicating that participants generally perceived themselves as having high levels of social support. The median score on the EPDS was 4, which was lower than the cutoff point for postpartum depression. The median total score on the DASS-21 was 4; median score on the

**Table 4. Descriptive statistics of breastfeeding problems and univariate analysis with breastfeeding self-efficacy (N = 787).**

| Breastfeeding problems variables | N (%) | Breastfeeding self-efficacy | | |
|---|---|---|---|---|
| | | Mean (SD) | t | P-Value |
| Breast pains | | | 1.69 | 0.091 |
| Yes | 469 (59.59) | 55.50 (8.79) | | |
| No | 318 (40.41) | 56.60 (9.07) | | |
| Back pains | | | 2.51 | 0.012* |
| Yes | 347 (44.09) | 55.05 (9.05) | | |
| No | 440 (55.91) | 56.65 (8.75) | | |
| Baby had trouble sucking or latching on onto the breast | | | 6.28 | < 0.001*** |
| Yes | 224 (28.46) | 52.86 (8.34) | | |
| No | 563 (71.54) | 57.17 (8.85) | | |
| Sore, cracked, or bleeding nipples | | | 1.62 | 0.106 |
| Yes | 335 (42.57) | 55.35 (8.80) | | |
| No | 452 (57.43) | 56.39 (8.98) | | |
| Not producing enough milk | | | 7.84 | < 0.001*** |
| Yes | 371 (47.14) | 53.40 (8.26) | | |
| No | 416 (52.86) | 58.21 (8.87) | | |
| C-Section affected breastfeeding | | | 3.15 | < 0.001*** |
| Yes | 135 (17.15) | 53.76 (8.86) | | |
| No | 652 (82.85) | 56.40 (8.87) | | |
| Episiotomy (cut vagina) | | | 2.18 | 0.030 |
| Yes | 43 (5.46) | 53.07 (7.91) | | |
| No | 744 (94.54) | 56.11 (8.95) | | |
| Doctor suggested not to breastfeed | | | 1.14 | 0.256 |
| Yes | 16 (2.03) | 53.44 (8.66) | | |
| No | 771 (97.97) | 56.00 (8.92) | | |
| Baby choked when breastfeeding | | | 0.42 | 0.678 |
| Yes | 484 (61.50) | 55.84 (8.93) | | |
| No | 303 (38.50) | 56.11 (8.90) | | |
| Baby wouldn't wake up to nurse regularly enough | | | 1.38 | 0.167 |
| Yes | 256 (32.53) | 55.31 (8.74) | | |
| No | 531 (67.47) | 56.25 (8.99) | | |
| Baby was not interested in nursing or got distracted | | | 3.20 | < 0.001*** |
| Yes | 127 (16.14) | 53.64 (8.63) | | |
| No | 660 (83.86) | 56.39 (8.91) | | |
| Baby nursed too often | | | 1.40 | 0.163 |
| Yes | 346 (43.96) | 55.45 (9.00) | | |
| No | 441 (56.04) | 56.34 (8.84) | | |
| Milk taking too long to secrete | | | 8.17 | < 0.001*** |
| Yes | 293 (37.23) | 52.71 (8.20) | | |
| No | 494 (62.77) | 57.87 (8.77) | | |
| Baby didn't gain enough weight or lost too much weight | | | 2.24 | 0.025* |
| Yes | 118 (14.99) | 54.25 (7.92) | | |
| No | 669 (85.01) | 56.24 (9.05) | | |
| Not enough time to feed child | | | 0.29 | 0.770 |
| Yes | 95 (12.07) | 55.69 (9.11) | | |
| No | 692 (87.93) | 55.98 (8.89) | | |
| Infection of the breasts (e.g., abscess, yeast) | | | 0.03 | 0.977 |

*(Continued)*

**Table 4.** (Continued)

| Breastfeeding problems variables | N (%) | Breastfeeding self-efficacy | | |
| --- | --- | --- | --- | --- |
| | | *Mean (SD)* | *t* | *P*-Value |
| Yes | 17 (2.16) | 55.88 (7.03) | | |
| No | 770 (97.84) | 55.95 (8.96) | | |
| Clogged milk duct | | | 2.72 | 0.007** |
| Yes | 134 (17.03) | 54.04 (9.32) | | |
| No | 653 (82.97) | 56.34 (8.79) | | |
| Breast engorgement | | | -0.43 | 0.665 |
| Yes | 529 (67.22) | 56.04 (8.81) | | |
| No | 258 (32.78) | 55.75 (9.15) | | |
| Milk leaked too much | | | -1.65 | 0.099 |
| Yes | 493 (62.64) | 56.35 (8.98) | | |
| No | 294 (37.36) | 55.27 (8.78) | | |

Notes.

* *P*< 0.05

**P< 0.01

***P<0.001.

t = value of t- test.

DASS-Depression, DASS-Anxiety, and DASS-Stress were 1,1, and 2, respectively; implying that the mental health status of mothers was normal in general.

The analysis finds that several the BSE Framework variables were significantly associated with the BSE of rural mothers (Table 5). In particular, breastfeeding attitudes of the mothers

**Table 5. Descriptive statistics of the Dennis breastfeeding self-efficacy framework variables excluding breastfeeding problems and univariate correlations with breastfeeding self-efficacy (N = 787).**

| Variables | Median (*IQR*) | Score range | The relationship with breastfeeding self-efficacy | |
| --- | --- | --- | --- | --- |
| | | | Spearman's correlation coefficient | *P*-Value |
| Breastfeeding Knowledge Score | 7 (5–8) | 2–11 | 0.051 | 0.151 |
| IIEAS Score | 62 (59–65) | 46–81 | 0.139 | < 0.001*** |
| EDPS Score | 4 (2–7) | 0–25 | -0.128 | < 0.001*** |
| DASS Score | 4 (1–9) | 0–46 | -0.161 | < 0.001*** |
| DASS of Depression | 1 (0–2) | 0–16 | -0.145 | < 0.001*** |
| DASS of Anxiety | 1 (0–2) | 0–16 | -0.098 | 0.006** |
| DASS of Stress | 2 (0–5) | 0–16 | -0.158 | < 0.001*** |
| Family Support Score for breastfeeding | 34 (30–36) | 15–45 | 0.310 | < 0.001*** |
| MSPSS Score | 66 (58–72) | 15–84 | 0.224 | < 0.001*** |
| Significant others | 23 (19–24) | 4–28 | 0.251 | < 0.001*** |
| Family | 23 (20–24) | 4–28 | 0.197 | < 0.001*** |
| Friends | 21 (18–24) | 4–28 | 0.136 | < 0.001*** |

Notes. Abbreviations: IIFAS, Lowa Infant Feeding Attitude; EDPS, Edinburgh Postnatal Depression; DASS, Scale of Depression Anxiety Stress; MSPSS, Multidimensional Scale of Perceived Social Support;

* P< 0.05

**P< 0.01

***P<0.001.

**Table 6. Determinants of breastfeeding self-efficacy among postpartum women in rural China, linear regression model (N = 787).**

| Variables | B | 95%CI | SE | β | t | P-Value |
|---|---|---|---|---|---|---|
| Breastfeeding attitudes [a] | 0.158 | (0.043, 0.272) | 0.058 | 0.088 | 2.71 | 0.007** |
| Social support from significant others [b] | 0.357 | (0.224,0.490) | 0.068 | 0.168 | 5.28 | < 0.001*** |
| Family support for breastfeeding | 0.433 | (0.307,0.560) | 0.065 | 0.219 | 6.71 | < 0.001*** |
| Baby had trouble sucking or latching on onto the breast | -3.361 | (-4.493, -2.143) | 0.620 | -0.170 | -5.42 | < 0.001*** |
| Not producing enough milk | -2.644 | (-3.903, -1.385) | 0.641 | -0.148 | -4.12 | < 0.001*** |
| Milk taking too long to secrete | -3.190 | (-4.493, -1.886) | 0.664 | -0.173 | -4.80 | < 0.001*** |

Notes. Adjusted $R^2$ = 0.24;

* $P < 0.05$

** $P < 0.01$

*** $P < 0.001$.

[a] Breastfeeding attitudes were assessed using the Iowa Infant Feeding Attitude Scale (IIFAS).

[b] Social support from significant others was assessed using the Multidimensional Scale of Perceived Social Support (MSPSS).

($r = 0.166$, $P < 0.001$), family support for breastfeeding ($r = 0.297$, $P < 0.001$) and social support ($r = 0.214$, $P < 0.001$) have significantly positive correlations with BSE. At the same time, the results show that depression (EPDS score: $r = -0.099$, $P = 0.006$; DASS of Depression: $r = -0.120$, $P < 0.001$) and stress ($r = -0.137$, $P < 0.001$) had significantly negative relationships with BSE.

## Identification of the determinants of BSE among postpartum women in rural China based on the Dennis BSE framework

The stepwise multiple linear regression finds that six variables explain 24% of the variance of BSE among postpartum women in rural China (Table 6). Breastfeeding attitudes ($\beta = 0.088$, $P < 0.001$), breastfeeding family support ($\beta = 0.168$, $P < 0.001$), and social support from significant others ($\beta = 0.219$, $P < 0.001$) were positively associated with BSE. In contrast, three breastfeeding problems, including the child having trouble sucking or latching onto the breast ($\beta = -0.170$, $P < 0.001$), not producing enough milk ($\beta = -0.148$, $P < 0.001$), and milk taking too long to secrete ($\beta = -0.173$, $P < 0.001$) were negatively associated with BSE (Table 6).

## Discussion

This study aimed to examine BSE and its determinants among postpartum women in rural China. To our knowledge, although previous research has studied the determinants of BSE in general, and among women living in urban China, this is the first study to explore the determinants of BSE in China's rural areas. This exploration may assist health care professionals in identifying mothers experiencing low BSE, who may be at risk of prematurely discontinuing breastfeeding, and identify possible target areas for researchers and practitioners seeking to improve BSE among women in rural China.

In our study, the overall average BSE score among the participants was 55.95, with a mean item score of 3.50. When we compare the mean item score of our study to samples obtained from cities in other regions of China, we find that the BSE scores of mothers in rural China are lower than those reported in Tianjin, Hongkong, and Shanghai, China. Previous studies in these three cities reported mean BSE item scores of 3.92, 3.54, and 3.67 [15, 16, 42], respectively. The difference in mean item scores can partially be explained by the fact that these three studies were all conducted in urban areas of China; the majority of the participants were

educated women and the population was likely to have (on average) a higher income, more family assets, and better support resources than individuals/families in rural China. Additionally, there is evidence that health care professionals in rural China often provide inadequate information/support on issues of child nutrition/breastfeeding, potentially causing further differences in BSE scores [43, 44].

Further examination of the BSE items in the present study suggests that mothers were least confident in their ability to exclusively breastfeed without their child receiving even some water. This lack of confidence in exclusive breastfeeding is largely consistent with the low rates of exclusive breastfeeding observed in other studies in rural China [13]. Moreover, compared to interpersonal concerns in breastfeeding, participants scored lower on items related to breastfeeding technique. These findings are similar to the study conducted in Xiamen, China, which also found that mothers were less confident in their breastfeeding technique [41]. Therefore, these results suggest that it may be necessary to develop interventions to educate women in breastfeeding techniques and promote women's confidence in exclusive breastfeeding in rural China.

The empirical results of this study also find that women who had negative breastfeeding experiences had significantly lower BSE than those without such experiences. Three variables related to breastfeeding problems, including the child having trouble sucking or latching onto the breast, not producing enough milk, and milk taking too long to secrete, were all significantly associated with lower BSE. Such breastfeeding problems may also explain the diminished confidence in breastfeeding techniques reported among postpartum women in our study. Although few studies have examined the role of negative breastfeeding experiences in BSE, the findings align with the Dennis BSE framework, which theorizes that successful performance accomplishments increase BSE, whereas repeated failures or difficulties diminish it [8].

Early challenges with breastfeeding may be particularly salient for BSE among postpartum mothers in rural China. Previous research has shown that more than half of postpartum women in rural China experienced problems in the early stages of breastfeeding [45]. In our study, 28.5% of postpartum women experienced difficulty with latching during the first two weeks of breastfeeding, 47.1% experienced insufficient milk supply, and 37.2% experienced slow milk secretion. Postpartum women who encounter these problems in the early stages of breastfeeding may feel inadequate in their breastfeeding techniques and overwhelmed by challenges, thus reducing BSE. Moreover, although these problems can be alleviated by educating women on effective breastfeeding techniques, it is often difficult for postpartum women in rural areas to obtain relevant counseling and guidance [46]. When breastfeeding problems arise but cannot be solved in a timely and effective manner, postpartum women's BSE decreases, and mothers may eventually give up breastfeeding [47]. Therefore, public health services in rural China should focus on helping new mothers resolve early problems they encounter during the breastfeeding process, especially insufficient milk, poor sucking or latching, and slow milk secretion.

In contrast to breastfeeding problems, the results find that social support from significant others and family support for breastfeeding were both significantly associated with higher BSE among postpartum women in rural China. This finding is consistent with BSE studies internationally [48, 49], as well as studies of self-efficacy in general, both of which find that social support can increase one's coping abilities and competence [50]. This also aligns with the Dennis BSE framework, which suggests that verbal persuasion from family members, especially significant others, encourages mothers to continue breastfeeding their infants despite challenges [8]. As the closest and most important social network, family members are particularly important sources of emotional support for postpartum women in general [51] and in breastfeeding promotion specifically [52]. In addition to emotional support, postpartum women with higher

levels of breastfeeding support receive relatively more practical assistance from family, which may help them to persist in breastfeeding [35]. In rural China, however, family members and significant others rarely receive education on breastfeeding or how to support breastfeeding mothers [41]. Educating family members about the importance of breastfeeding support for postpartum women may therefore increase BSE, motivation to breastfeed, and success in breastfeeding.

Consistent with previous research [49], positive breastfeeding attitudes among mothers were also found to be significantly associated with higher BSE in our study. Attitudes towards breastfeeding have also been identified as an indicator of breastfeeding behavior among women in urban China [33]. Encouraging postpartum women to develop positive attitudes towards breastfeeding may improve BSE and promote breastfeeding among women in rural China.

Finally, our study found that physiological or emotional responses, including stress, anxiety, and depression, were not determinants of BSE among women in rural China. This contradicts to the Dennis BSE framework, which posits that physiological or emotional responses can affect BSE [8]. These findings also contradict a previous study in Vietnam, which found that mothers with a higher level of postpartum depression tend to have lower BSE in the early postpartum period [10]. This discrepancy may be due to cultural differences between rural China and Vietnam, and further research is needed to better understand the links between mental health and BSE.

## Limitations

Some important factors such as prior breastfeeding experiences (a component of performance accomplishments in the Dennis BSE framework) and effects of role modeling (a component of vicarious experience in the Dennis BSE framework) were not collected in this study and should be examined in further research. Furthermore, as our assessment of BSE was collected at a single point in time, we were unable to examine how BSE may change over the duration of breastfeeding, and we may have missed determinants of BSE that evolve with maternal breastfeeding experience. Further research is needed to examine the full range of determinants of BSE and their temporal and causal associations to BSE, to help health care professionals identify mothers at different stages of the postpartum period who are at high risk of low BSE and develop effective interventions to improve BSE in rural China.

## Implications

This study highlights the importance of improving BSE, and particularly self-efficacy in exclusive breastfeeding, among postpartum women in rural China by identifying some of the primary determinants of BSE. Health care professionals should develop multi-dimensional strategies to foster BSE, such as intervening to enhance the breastfeeding attitudes of mothers, adopting a family-centered approach in the provision of breastfeeding education, and rallying comprehensive social support for postpartum women. The findings of our study also indicate that health care providers should increase education on breastfeeding techniques and assist women in resolving common breastfeeding problems, such as poor latching, insufficient breast milk and slow secretion of breast milk, in order to improve BSE.

## Conclusions

The findings indicate that BSE among postpartum women in rural China is relatively low compared to urban China, pointing to a need for strategies to promote BSE. Positive attitudes towards breastfeeding, as well as social support and family support for breastfeeding,

contribute to greater BSE in rural China. In contrast, difficulties with breastfeeding are associated with reduced BSE. Researchers and practitioners should investigate effective strategies to improve social support for breastfeeding, promote positive attitudes towards breastfeeding, and provide women with education on breastfeeding techniques and actionable solutions to breastfeeding problems. With greater effort placed on these now-identified critical points, BSE and breastfeeding practices could be meaningfully improved in rural China.

## Supporting information

**S1 Table. 19 items of the Breastfeeding Problems Questionnaire.**
(DOCX)

**S2 Table. 12 items of the Breastfeeding Knowledge Questionnaire.**
(DOCX)

**S1 Dataset. Stata data files.**
(DTA)

## Acknowledgments

The authors would like to thank the local officials in Sichuan Province for their collaboration, as well as all the postpartum mothers from each of the participating sites in this study for providing substantial assistance towards the collection of our survey data.

## Author Contributions

**Conceptualization:** Linhua Li, Yuju Wu, Alexis Medina, Scott Rozelle, Huan Zhou.

**Formal analysis:** Linhua Li, Yuju Wu.

**Funding acquisition:** Huan Zhou.

**Investigation:** Linhua Li, Yuju Wu, Qingzhi Wang, Yefan Du.

**Methodology:** Linhua Li, Yuju Wu, Qingzhi Wang, Yefan Du, Dimitris Friesen, Yian Guo, Alexis Medina, Scott Rozelle.

**Project administration:** Qingzhi Wang, Sarah-Eve Dill, Alexis Medina, Scott Rozelle, Huan Zhou.

**Resources:** Alexis Medina, Scott Rozelle, Huan Zhou.

**Supervision:** Huan Zhou.

**Writing – original draft:** Linhua Li, Yuju Wu.

**Writing – review & editing:** Linhua Li, Yuju Wu, Qingzhi Wang, Yefan Du, Dimitris Friesen, Yian Guo, Sarah-Eve Dill, Scott Rozelle, Huan Zhou.

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
