## [Decision Letter · Decision Letter 0]

28 Jan 2022

PONE-D-21-31115Predictors of breastfeeding self-efficacy among postpartum women in rural China: A cross-sectional studyPLOS ONE

Dear Dr. Huan Zhou,

Thank you for submitting your manuscript to PLOS ONE. After careful consideration, we feel that it has merit but does not fully meet PLOS ONE’s publication criteria as it currently stands. Therefore, we invite you to submit a revised version of the manuscript that addresses the points raised during the review process.

ACADEMIC EDITOR:

Dear authors on your scholarly work; you have brought an important study. However, the manuscript has some language usage flaws including punctuations, wordings, spelling and grammar errors. These problems are found throughout the manuscript. Moreover, there are several methodological limitations as the reviewer raised. Therefore, kindly address all the reviewers’ concerns and also please make repeated proof-reading before resubmitting the manuscript. This would help increase the readership of the manuscript if published.

We look forward to receiving your revised manuscript.

Kind regards,

Wubet Alebachew Bayih, M.Sc.

Academic Editor

PLOS ONE

Journal Requirements:

Reviewers' comments:

Reviewer's Responses to Questions

**Comments to the Author**

1. Is the manuscript technically sound, and do the data support the conclusions?

Reviewer #1: Partly

Reviewer #2: Yes

2. Has the statistical analysis been performed appropriately and rigorously? 

Reviewer #1: Yes

Reviewer #2: Yes

3. Have the authors made all data underlying the findings in their manuscript fully available?

Reviewer #1: Yes

Reviewer #2: Yes

4. Is the manuscript presented in an intelligible fashion and written in standard English?

Reviewer #1: No

Reviewer #2: No

5. Review Comments to the Author

Reviewer #1: Predictors of breastfeeding self-efficacy among postpartum women in rural China: A cross-sectional study

First I want to thank you for the chance to review this interesting manuscript entitled “Predictors of breastfeeding self-efficacy among postpartum women in rural China: A cross-sectional study”. The study presented in this manuscript is on an important and timely topic. It presents pertinent information for the audience of this journal and would be of interest to its readership. I will first make some general comments and then add specific areas under each section suggested for revision.

General Comments

Before publication, the manuscript will need clearness in the methods and results sections. The authors could use the assistance of an editor for basic grammar and sentence structure corrections as well.

PLOS ONE Comments to the Author

1. Is the manuscript technically sound, and do the data support the conclusions?

Partly

2. Has the statistical analysis been performed appropriately and rigorously?

Yes

3. Have the authors made all data underlying the findings in their manuscript fully available?

Yes

4. Is the manuscript presented in an intelligible fashion and written in Standard English?

No

5. Review Comments to the Author

The manuscript is good. However, it does need an edition by a native English speaker or Journal English language services.

6. PLOS authors have the option to publish the peer review history of their article

Do you want your identity to be public for this peer review?

Yes, Hassen Mosa

Introduction: Several sections of the introduction need an attention. Readers will expect to see in brief, the literature gap that authors wanted to fill. Additionally, please rewrite and modify your introduction part by removing the frequently misused words.

Methods: The method part needs more amendment. Particularly, the measurement part should be revised and shortened for more simplicity. Regarding the data collection, you have mention whether questionnaire is adapted or adopted, and say something about the validity and reliability of your tool.

Conclusion: Your conclusion is somewhat vague. Better if you write it clearly according to your finding.

References

Please, review your references and adjust according to the PLOS specifications.

Since PLOS ONE does not copyedit accepted manuscripts, the authors should employ an editor to assist with ambiguous and grammatical errors that appear throughout the text. There are multiple grammar and sentence structure corrections that are required prior to publication.

Reviewer #2: Predictors of breastfeeding self-efficacy among postpartum women in rural China: A cross-sectional study

Dear Authors,

Thank you for the chance to review the manuscript. My detailed remarks on the review can be found here.

In the title should be used an appropriate word/term in line with epidemiological study deign, which is predictors leads to case-control rather than cross-sectional study. Please modify it?

In the abstract section line 37-39, all of the factors inversely associated with the predictors of BSE, but we are interpreting with a wrong ways. Please see and correcte accordingly.

Your conclusion should be consider drawing out the “so whats” of your findings to drive the points, but we are stated your conclusion that are the roles of health professionals.

Introduction; Your manuscript missing a section which generally describes the breastfeeding self-efficacy among postpartum women landscapes in China. What is breastfeeding self-efficacy? Why suggests breastfeeding self-efficacy among postpartum women? Has breastfeeding self-efficacy helped elsewhere? What is the general breastfeeding self-efficacy rate? All these you can touch on briefly?

Method; line 123-124: As stated above when use the term ‘predictors’ the design should be case-control.

Sample; line 139: why we use survey for this particular study, and also how to selected the actual participant of the study. Please elaborate it?

Line 141-150: These sentences are unclear. Kindly revisit.

Line 181: I am not sure you can use the word “extensive” as your manuscript has not really demonstrated the “extensiveness” of your approach.

Line 268; ‘November to December, 2019’ are you sure this was an enough time to conducted enumeration based on your study area coverage? Please see again and modify accordingly.

Line 273; why we used only 20 participants as pilot study and what was your reference’s we used only 20 participants?

Age category; which age classification are used in the study, which was ’18-30 Vs >31’?

Line 305-306 and line 315-316; we used mean score ± standard deviation, sure that the data distribution is normal?

Discussion: your discussion could be further developed so it doesn’t appear like a repetition of your objectives.

Line 464-466; this is the limitation of the study design itself not yours, please modify it. Why we are incorporating your strength?

General comment; There are several grammatical issues in manuscript which make it difficult for the readers to grasp the important points to convey in sections where they occur. I suggest you critically review the manuscript or possibly the service of copyeditor.

Thanks!!!

6. PLOS authors have the option to publish the peer review history of their article (what does this mean?). If published, this will include your full peer review and any attached files.

Reviewer #1: **Yes: **Hassen Mosa

Reviewer #2: No

---

## [Author Response · Author response to Decision Letter 0]

13 Mar 2022

Dear Editor,

My coauthors and I thank you for your comments and suggestions concerning our manuscript “Determinants of breastfeeding self-efficacy among postpartum women in rural China: A cross-sectional study” (No: PONE-D-21-31115). We also deeply appreciate the thoughtful comments from the reviewers, which have improved the paper. 

We have studied the comments carefully and have revised our paper accordingly. This letter provides point-by-point responses to each comment and summarizes relevant changes in the manuscript. These changes are highlighted yellow in the revised manuscript. Primary changes to the paper include the following:

• We have revised the title of the manuscript to “Determinants of breastfeeding self-efficacy among postpartum women in rural China: A cross-sectional study” and replaced “predictors” with “determinants” throughout the revised manuscript.

• We have reworked the Introduction section to ensure a brief literature gap and bring the landscapes of Chinese postpartum women’s breastfeeding self-efficacy.

• We have updated a detailed description of the process of selecting the actual participants of the study in the Methods section.

• We have reworked the Conclusion section to make it clearly according to our findings. 

• We have also obtained the assistance of an editor to correct language usage flaws including punctuations, wordings, spelling and grammar errors throughout the manuscript.

The material in the manuscript has not and will not be offered elsewhere for possible publication as long as it is under consideration by PLOS ONE.

Once again, we are very grateful for your consideration of our work. We look forward to receiving your feedback on the updated manuscript.

With best regards,

Huan Zhou

Professor, Chair, Department of Health Behavior and Social Medicine

West China School of Public Health, Sichuan University

Email: zhouhuan@scu.edu.cn

 

Editor’s comments to author: 

Editor Comment: Dear authors on your scholarly work; you have brought an important study. However, the manuscript has some language usage flaws including punctuations, wordings, spelling and grammar errors. These problems are found throughout the manuscript. Moreover, there are several methodological limitations as the reviewer raised. Therefore, kindly address all the reviewers’ concerns and also please make repeated proof-reading before resubmitting the manuscript. This would help increase the readership of the manuscript if published.

Response to Editor Comment: 

Thank you for your time and consideration. We have addressed each reviewer comment and have revised the manuscript according to the reviewers’ comments. Specifically, we have:

• Revised the title of the manuscript to “Determinants of breastfeeding self-efficacy among postpartum women in rural China: A cross-sectional study” and replaced “predictors” with “determinants” throughout the revised manuscript;

• Reworked the Introduction section to include a brief literature review on the current research landscape of women’s breastfeeding self-efficacy in China and more clearly identify gaps in the literature;

• Added a detailed description of our process for selecting study participants in the Methods section;

• Reworked the Conclusion section to speak more closely to the specific findings of our study.;

• Obtained the assistance of an editor to correct language usage flaws including punctuations, wordings, spelling and grammar errors throughout the manuscript.

In the following pages, we provide a point-by-point response to the reviewers’ comments, including specific changes made to the revised manuscript. All changes are highlighted yellow in the revised manuscript.

 

Response to Reviewer 1

General Comment1: First I want to thank you for the chance to review this interesting manuscript entitled “Predictors of breastfeeding self-efficacy among postpartum women in rural China: A cross-sectional study”. The study presented in this manuscript is on an important and timely topic. It presents pertinent information for the audience of this journal and would be of interest to its readership. I will first make some general comments and then add specific areas under each section suggested for revision.

Response to General Comment: 

Thank you for your comments and recognizing the value of our research. We have carefully considered your comments and revised our paper accordingly.

General Comment2: Before publication, the manuscript will need clearness in the methods and results sections. The authors could use the assistance of an editor for basic grammar and sentence structure corrections as well.

Response to General Comment: 

Thank you for pointing this out. We have reworked the Methods and Results sections to ensure clearness (for specific changes, please see our responses to comment 2 in “Response to Reviewer 1” and comments 5-12 in “Response to Reviewer 2”). We have also obtained the assistance of an editor to correct errors in grammar and sentence structure throughout the manuscript. Every change in the manuscript has been highlighted in yellow.

Comment 1: Introduction: Several sections of the introduction need an attention. Readers will expect to see in brief, the literature gap that authors wanted to fill. Additionally, please rewrite and modify your introduction part by removing the frequently misused words.

Response: Thank you for pointing this out. We have modified the Introduction part to remove misused words. All changes are highlighted yellow in the revised manuscript.

We have also clarified the gaps in the existing literature that our study seeks to fill in Introduction section of the revised manuscript on page 6-8, lines 111-166 (revised text in italics):

“Empirical studies have found evidence linking factors such as positive breastfeeding experiences, breastfeeding knowledge, breastfeeding attitude, social support, and postpartum depression, to BSE [10,11]. However, negative performance accomplishments (i.e., difficulties in successfully breastfeeding) have been less studied internationally, leaving a gap to be filled in the literature.

In addition, because cultural context may influence self-efficacy and related factors, there is a need for more studies of BSE in various LMIC settings, particularly those with documented low rates of breastfeeding. One understudied setting with low rates of breastfeeding is rural China. China is the most populous country in the world, and more than 60% of the country’s population lives in rural areas. Although the weighted prevalence for breastfeeding in China is 79.6%, only 20.8% of infants are breastfed exclusively for six months [12]. In rural China, exclusive breastfeeding rates among children younger than six months are even lower: a survey in 26 poor, rural counties of China found that the rate of exclusive breastfeeding was only 58.3% among newborn infants, declining further to 29.1% in those aged three to four months and 13.6% in those aged five to six months [13].

Little is known about BSE in rural China; however, previous studies have measured the level of BSE among postpartum women in urban areas of China. These studies have found that the mean scores of items on the Breastfeeding Self-Efficacy Scale in Hong Kong, Shanghai and Guangzhou were 3.92 [14], 3.67 [15] and 3.38 [16], respectively. Interventions targeting BSE have also been shown to be effective in promoting breastfeeding practices among postpartum women in urban areas in China [17,18]. Because self-efficacy is a psychological category that is related to many socio-cultural factors, however, the results of international studies and studies in China’s urban areas are not necessarily representative of rural China, and the lack of studies examining BSE among postpartum women in rural China presents another gap in the existing literature. Given the low rates of breastfeeding in rural China and the important role that BSE plays in breastfeeding behavior in the international literature, research on BSE in rural China is needed to inform public health policies and improve breastfeeding outcomes. Therefore, this study aims to describe the BSE of postpartum women in rural China, and to identify the determinants of BSE among postpartum women in rural China based on the Dennis BSE framework.”

REFERENCES:

14. Ip WY, Gao LL, Choi KC, Chau JPC, Xiao Y. The Short Form of the Breastfeeding Self-Efficacy Scale as a Prognostic Factor of Exclusive Breastfeeding among Mandarin-Speaking Chinese Mothers. J Hum Lact. 2016;32(4):711–20. PMID: 27474407

15. Li T, Guo N, Jiang H, Eldadah M. Breastfeeding Self-Efficacy Among Parturient Women in Shanghai: A Cross-Sectional Study. J Hum Lact. 2019 Aug 5;35(3):583–91. http://journals.sagepub.com/doi/10.1177/0890334418812044 PMID: 30517822

16. Dai X, Dennis CL. Translation and validation of the breastfeeding Self-Efficacy Scale into Chinese. J Midwifery Women’s Heal. 2003;48(5):350–6. https://doi.org/10.1016/s1526-9523(03)00283-6 PMID: 14526349

17. Wu DS, Hu J, Mccoy TP, Efird JT. The effects of a breastfeeding self-efficacy intervention on short-term breastfeeding outcomes among primiparous mothers in Wuhan, China. J Adv Nurs. 2014;70(8):1867–79. https://doi.org/10.1111/jan.12349 PMID: 24400967

18. Liu Y, Li N, Mei Z, Li Z, Ye R, Zhang L, et al. Effects of prenatal micronutrients supplementation timing on pregnancy-induced hypertension: Secondary analysis of a double-blind randomized controlled trial. Matern Child Nutr. 2021;(January):e13157. https://doi.org/10.1111/mcn.13157 PMID: 33594802

Comment 2: Methods: The method part needs more amendment. Particularly, the measurement part should be revised and shortened for more simplicity. Regarding the data collection, you have mention whether questionnaire is adapted or adopted, and say something about the validity and reliability of your tool.

Response: Thank you for bringing this to our attention. We have shortened the measurement part of the methods and added details on the validity and reliability of each tool in the Methods section of the revised manuscript on page 11-15, lines 226-328 (revised text in italics). If the reviewer still feels that this section is too long, we are happy to move details on the measurements to a supplemental appendix.

“The “BSE” outcome was measured using the Chinese (Mandarin) version of the Breastfeeding Self-Efficacy Scale-Short Form (BSES-SF), which has been validated in mainland China with a Cronbach’s alpha of 0.94 [5]. The BSES-SF is comprised of 14 positively-worded statements regarding mothers’ self-efficacy in their ability to breastfeed [16, 21]. In addition to these 14 items, we also included two items (“I can always exclusively breastfeed without my child receiving even a drop of water;” and “I can always stop someone from trying to feed my child liquids or foods other than breast milk before six months of age”) adapted from a BSE scale by Boateng et al. [22] to better measure maternal self-efficacy in exclusive breastfeeding in rural China. Two experienced independent researchers fluent in English and Chinese translated the two additional items into Chinese (Mandarin) before addition. Combining the 14 items from the BSES-SF with the two items from Boateng et al [22], the BSE questionnaire in this study includes 16 items measured on a 5-point Likert-type scale with answers ranging from not at all confident (1) to always confident (5). Responses were summed to calculate a total score ranging from 16 to 80, with higher scores indicating higher BSE. In the current study, the Cronbach’s alpha for this questionnaire is 0.88.”

After an extensive literature review, the research team developed a “Breastfeeding Problems Questionnaire” that encompasses the most prevalent problems associated with breastfeeding [23-31]. This questionnaire contains 19 items that measures concerns or difficulties of mothers during the first two weeks of breastfeeding. Each question in the questionnaire was answered with a “yes” or “no” answer. All items of the Breastfeeding Problems Questionnaire are presented in S1 Table.

Breastfeeding attitudes were assessed using the Iowa Infant Feeding Attitude Scale (IIFAS) [32]. This 17-item scale covers various dimensions of infant feeding attitudes, which mothers were asked to rank on a 5-point Likert scale from strongly disagree (1) to strongly agree (5). Higher scores indicate a more positive attitude to breastfeeding. The tool has been found to be reliable and valid, with a Cronbach’s alpha of 0.62 in mainland China [27]. In the present study, the Cronbach’s alpha is 0.56.

The “Breastfeeding Knowledge Questionnaire” was adapted and modified from the Breastfeeding Knowledge Questionnaire-Short Form (BFKQ-SF) [33] by the research team to fit the setting of rural China. This questionnaire has 12 items. Each correct answer is scored as 1, while wrong or unclear answers are scored as 0. The total score ranges from 0-12, with higher scores indicating greater knowledge about breastfeeding. All items of the Breastfeeding Knowledge Questionnaire can be found in S2 Table.

Family support for breastfeeding perceived by the mother was measured using a scale designed by Zhu et al. [34]. The scale contains nine items, with response ranked on a Likert scale ranging from strongly disagree (1) to strongly agree (4). Mothers with scores of 27 or higher are considered to have positive support. The scale has been evaluated for reliability and validity and has been proven to be effective at measuring family support for the breastfeeding of mothers [29]. In the current study, the Cronbach’s alpha is 0.78.

The Multidimensional Scale of Perceived Social Support (MSPSS) was used to measure perceived social support (unrelated to breastfeeding) from family, friends and significant others [35]. This scale contains 12 items, with responses ranked on a 7-point Likert scale from strongly disagree (1) to strongly agree (7). Scores range from 12 to 84, with higher scores indicating higher levels of perceived social support [35]. In the present study, the Cronbach’s alphas for the MSPSS total scale and family, friends, and significant other subscales are 0.89, 0.82, 0.85, and 0.80, respectively.

The 21-item version of the Depression Anxiety Stress Scales (DASS-21) is a 21-item questionnaire first presented by Lovibond in 1995 that uses seven questions to measure each of the symptoms of stress, anxiety, and depression [36]. This questionnaire is designed as a Likert questionnaire, with item scores ranging from zero to three indicating different levels of severity of a particular symptom experienced over the past week. In the present study, the Cronbach’s alpha for the DASS-21 total scale and depression, anxiety, and stress subscales are 0.91, 0.82, 0.71, and 0.81, respectively. 

The Edinburgh Postnatal Depression Scale (EPDS) is a 10-item instrument developed to identify mothers who may be experiencing postpartum depression [37]. Each item has four possible answers, with item scores ranging from zero to three. Total possible scores range from 0 to 30, with higher scores indicating a more elevated risk for postpartum depression [38]. The cutoff point for assessing depression varies by country, with an appropriate EPDS cutoff score of >10 for postnatal depression in China. In the current study, the Cronbach’s alpha is 0.79.”

Comment 3: Conclusion: Your conclusion is somewhat vague. Better if you write it clearly according to your finding.

Response: Thank you for pointing this out. We have updated the sentences in the Conclusion section on page 39, line 659-674 of the revised manuscript (revised text in italics):

“The findings indicate that BSE among postpartum women in rural China is relatively low compared to urban China, pointing to a need for strategies to promote BSE. Positive attitudes towards breastfeeding, as well as social support and family support for breastfeeding, contribute to greater BSE in rural China. In contrast, difficulties with breastfeeding are associated with reduced BSE. Researchers and practitioners should investigate effective strategies to improve social support for breastfeeding, promote positive attitudes towards breastfeeding, and provide women with education on breastfeeding techniques and actionable solutions to breastfeeding problems. With greater effort placed on these now-identified critical points, BSE and breastfeeding practices could be meaningfully improved in rural China.”

Comment 4: References

Please, review your references and adjust according to the PLOS specifications.

Response: We have reviewed and adjusted our references to ensure that they are in compliance with PLOS specifications. All changes are highlighted yellow in the revised manuscript.

Comment 5: Since PLOS ONE does not copyedit accepted manuscripts, the authors should employ an editor to assist with ambiguous and grammatical errors that appear throughout the text. There are multiple grammar and sentence structure corrections that are required prior to publication.

Response: Thank you again for bringing this to our attention. We have also obtained the assistance of an editor to correct errors in grammar and sentence structure throughout the manuscript. Every change in the manuscript has been highlighted in yellow.

 

Response to Reviewer 2

Comment 1：In the title should be used an appropriate word/term in line with epidemiological study deign, which is predictors leads to case-control rather than cross-sectional study. Please modify it?

Response: Thank you for pointing this out. This study is a cross-sectional study, and the appropriate term is “determinants” rather than “predictors”. We have revised the title of the manuscript to read as follows (revised text in italics)：

“Determinants of breastfeeding self-efficacy among postpartum women in rural China: A cross-sectional study”

Comment 2：In the abstract section line 37-39, all of the factors inversely associated with the predictors of BSE, but we are interpreting with a wrong way. Please see and correct accordingly.

Response: In our study, stepwise multiple linear regression analysis was used to explore the factors influencing BSE. The results of the regression found that six variables explained 24% of the variance in BSE. The six variables include positive breastfeeding attitudes, social support from significant others, family support for breastfeeding, the child having trouble sucking or latching onto the breast, the mother not producing enough milk and the mother’s milk taking too long to secrete. 

Among these variables, the regression coefficients for breastfeeding attitudes (β= 0.088, P< 0.001), breastfeeding family support (β= 0.168, P< 0.001), and social support from significant others (β= 0.219, P< 0.001), were positive, indicating that higher scores on these variables were associated with higher BSE. In contrast the regression coefficients of the three variables related to breastfeeding problems, including the child having trouble sucking or latching onto the breast (β= -0.170, P< 0.001), not producing enough milk (β= -0.148, P< 0.001) and milk taking too long to secrete (β= -0.173, P< 0.001), were negative, indicating that difficulties with breastfeeding were associated with reduced BSE. We have updated the sentence in the Abstract section of the revised manuscript page 2-3, lines 36-43 (revised text in italics):

“Participants reported a moderate level of breastfeeding self-efficacy, with an item mean score of 3.50. Self-efficacy was lowest for exclusive breastfeeding. Breastfeeding attitudes (β= 0.088, P< 0.001), breastfeeding family support (β= 0.168, P< 0.001), and social support from significant others (β= 0.219, P< 0.001) were positively associated with breastfeeding self-efficacy. Breastfeeding problems, including trouble with latching (β= -0.170, P< 0.001), not producing enough milk (β= -0.148, P< 0.001), and milk taking too long to secrete (β= -0.173, P< 0.001) were negatively associated with breastfeeding self-efficacy.”

Comment 3: Your conclusion should be consider drawing out the “so whats” of your findings to drive the points, but we are stated your conclusion that are the roles of health professionals.

Response: Thank you for bringing this to our attention. We have updated the sentences in the Conclusion section on page 39, line 659-674 of the revised manuscript (revised text in italics):

“The findings indicate that BSE among postpartum women in rural China is relatively low compared to urban China, pointing to a need for strategies to promote BSE. Positive attitudes towards breastfeeding, as well as social support and family support for breastfeeding, contribute to greater BSE in rural China. In contrast, difficulties with breastfeeding are associated with reduced BSE. Researchers and practitioners should investigate effective strategies to improve social support for breastfeeding, promote positive attitudes towards breastfeeding, and provide women with education on breastfeeding techniques and actionable solutions to breastfeeding problems. With greater effort placed on these now-identified critical points, BSE and breastfeeding practices could be meaningfully improved in rural China.”

Comment 4: Introduction; Your manuscript missing a section which generally describes the breastfeeding self-efficacy among postpartum women landscapes in China. What is breastfeeding self-efficacy? Why suggests breastfeeding self-efficacy among postpartum women? Has breastfeeding self-efficacy helped elsewhere? What is the general breastfeeding self-efficacy rate? All these you can touch on briefly?

Response: Thank you for pointing this out. The concept of breastfeeding self-efficacy was developed by Dennis and refers to the level of confidence mothers have in their ability to breastfeed their babies. Breastfeeding self-efficacy plays an important role in postpartum women’s breastfeeding behavior. Interventions targeting breastfeeding self-efficacy have proven to be effective in promoting breastfeeding practices among postpartum women in urban areas in China. Studies have measured the level of breastfeeding self-efficacy among postpartum women in urban areas of China, and the results showed that the mean scores of items on the Breastfeeding Self-Efficacy Scale in Hong Kong, Shanghai and Guangzhou were: 3.92, 3.67 and 3.38 respectively. 

We have added literature on breastfeeding self-efficacy internationally to the Introduction section of the revised manuscript page 4-5, lines 77-88 (revised text in italics):

“The international literature has shown that breastfeeding self-efficacy (BSE) is one of the most crucial, modifiable factors influencing postpartum women’s breastfeeding behavior [5,6]. BSE is derived from the self-efficacy concept of Bandura [7]. Dennis developed a framework for BSE 1999 [8], defining BSE as a mother’s perceived ability to breastfeed her child. In Dennis’ BSE framework, BSE influences a mother’s breastfeeding decisions, including the decision to breastfeed, how much effort should be given to breastfeeding, and how to respond to challenges during breastfeeding [5,6,8]. High BSE has also been associated with outcomes such as exclusive breastfeeding among postpartum women in the first 6 months after birth [9].”

We have described the landscape of Chinese postpartum women’s breastfeeding self-efficacy to the Introduction section of the revised manuscript page 7-8, lines 142-155 (revised text in italics):

Little is known about BSE in rural China; however, previous studies have measured the level of BSE among postpartum women in urban areas of China. These studies have found that the mean scores of items on the Breastfeeding Self-Efficacy Scale in Hong Kong, Shanghai and Guangzhou were 3.92 [14], 3.67 [15] and 3.38, [16] respectively. Interventions targeting BSE have also been shown to be effective in promoting breastfeeding practices among postpartum women in urban areas in China [17,18]. Self-efficacy is a psychological category that is related to many socio-cultural factors, and the results of foreign and other domestic studies cannot be directly used to guide the practice in rural China. The lack of studies examining BSE among postpartum women in rural China presents another gap in the existing literature. Given the low rates of breastfeeding in rural China and the important role that BSE plays in breastfeeding behavior in the international literature, research on BSE in rural China is needed to inform public health policies and improve breastfeeding outcomes.”

REFERENCES:

5. Glassman ME, McKearney K, Saslaw M, Sirota DR. Impact of breastfeeding self-efficacy and sociocultural factors on early breastfeeding in an urban, predominantly dominican community. Breastfeed Med. 2014;9(6):301–7. https://doi.org/10.1089/bfm.2014.0015 PMID: 24902047

6. Lau CYK, Lok KYW, Tarrant M. Breastfeeding Duration and the Theory of Planned Behavior and Breastfeeding Self-Efficacy Framework: A Systematic Review of Observational Studies. Matern Child Health J. 2018;22(3):327–42. http://dx.doi.org/10.1007/s10995-018-2453-x PMID: 29427014

7. Bandura A. Self-efficacy: Toward a unifying theory of behavioral change. Psychol Rev. 1977;84(2):191–215. http://doi.apa.org/getdoi.cfm?doi=10.1037/0033-295X.84.2.191 PMID: 847061

8. Dennis C-L. Theoretical Underpinnings of Breastfeeding Confidence: A Self-Efficacy Framework. J Hum Lact. 1999 Sep;15(3):195–201. http://journals.sagepub.com/doi/10.1177/089033449901500303 PMID: 10578797

9. Jama NA, Wilford A, Masango Z, Haskins L, Coutsoudis A, Spies L, et al. Enablers and barriers to success among mothers planning to exclusively breastfeed for six months: A qualitative prospective cohort study in KwaZulu-Natal, South Africa. Int Breastfeed J. 2017;12(1):1–13. https://doi.org/10.1186/s13006-017-0135-8 PMID: 29026431

14. Ip WY, Gao LL, Choi KC, Chau JPC, Xiao Y. The Short Form of the Breastfeeding Self-Efficacy Scale as a Prognostic Factor of Exclusive Breastfeeding among Mandarin-Speaking Chinese Mothers. J Hum Lact. 2016;32(4):711–20. PMID: 27474407

15. Li T, Guo N, Jiang H, Eldadah M. Breastfeeding Self-Efficacy Among Parturient Women in Shanghai: A Cross-Sectional Study. J Hum Lact. 2019 Aug 5;35(3):583–91. http://journals.sagepub.com/doi/10.1177/0890334418812044 PMID: 30517822

16. Dai X, Dennis CL. Translation and validation of the breastfeeding Self-Efficacy Scale into Chinese. J Midwifery Women’s Heal. 2003;48(5):350–6. https://doi.org/10.1016/s1526-9523(03)00283-6 PMID: 14526349

17. Wu DS, Hu J, Mccoy TP, Efird JT. The effects of a breastfeeding self-efficacy intervention on short-term breastfeeding outcomes among primiparous mothers in Wuhan, China. J Adv Nurs. 2014;70(8):1867–79. https://doi.org/10.1111/jan.12349 PMID: 24400967

18. Liu Y, Li N, Mei Z, Li Z, Ye R, Zhang L, et al. Effects of prenatal micronutrients supplementation timing on pregnancy-induced hypertension: Secondary analysis of a double-blind randomized controlled trial. Matern Child Nutr. 2021;(January):e13157. https://doi.org/10.1111/mcn.13157 PMID: 33594802

Comment 5: Method; line 123-124: As stated above when use the term ‘predictors’ the design should be case-control.

Response: Thank you again for raising this point. As mentioned above, our study is a cross-sectional descriptive study. We incorrectly used the term “predictors”. We have replaced “predictors” with “determinants” throughout the revised manuscript and highlighted all revisions. 

Comment 6: Sample; line 139: why we use survey for this particular study, and also how to select the actual participant of the study. Please elaborate it?

Response: In this comment, the reviewer makes two points. The first point asks why we use a survey for this particular study. The second point asks about how we select the study participants of the study. For clarity, we will respond to each point separately.

In response to the first point:

This study used a large-scale cross-sectional survey to collect data. This research design was based on two main considerations. First, this study focuses on understanding the status of breastfeeding self-efficacy among postpartum women in rural China. A large-scale survey gives the research team adequate statistical power to identify population-level trends and correlations. Second, as the target population of this study is postpartum women in rural areas, the survey research method can help us to obtain first-hand research data directly from postpartum women in rural areas of China. 

We have added this information to the Research Design subsection of the Methods section on page 9, lines 169-177 in the revised manuscript (revised text in italics):

“This study was conducted using a using a multi-stage random cluster sampling design and cross-sectional survey to assess BSE among women 0-6 months postpartum in rural China and to identify determinants of BSE based on the Dennis BSE framework. This research design was based on two main considerations. First, a large-scale survey gives the research team adequate statistical power to identify population-level trends and correlations. Second, the survey research method can help us to obtain first-hand research data directly from postpartum women in rural areas of China.”

In response to the second point:

The research team implemented a three-step sampling protocol to select the participants for the study. First, four nationally-designated poverty counties were selected within the sample prefecture. Second, sample townships were chosen within each sample county. To select townships representative of typical rural areas, the sampling frame excluded non-rural townships and rural townships with populations of less than 10,000. Of the remaining townships, 20 townships per county were randomly selected by a computer-generated random numbers method, resulting in a total of 80 townships. Finally, a list of all mothers with registered births within the past 6 months of the survey was obtained from the township health center in each sample township. A total of 842 postpartum women were identified and contacted by the research team, with the assistance of local township health center doctors or village doctors to invite.

In total, 55 eligible postpartum women failed to enroll in the study due to out-migration or travel at the time of the survey, intellectual disability or mental illness that prevented ability to give informed consent, or refusal to participate. Of the 842 lactating postpartum women who enrolled in this study, 787 postpartum women completed all aspects of the questionnaire, a response rate of 93.5%.

We have updated a detailed description of the process of selecting the actual participants of the study in the Methods section of the paper, page 10-11, lines 192-208 in the revised manuscript (revised text in italics):

“The research team sampled rural mothers within the 0-6 months postpartum period living in one prefecture of Sichuan province, China, following a multi-stage cluster sampling protocol. First, four nationally-designated poverty counties were selected within the sample prefecture. Second, sample townships were chosen within each sample county. To select townships representative of typical rural areas, the sampling frame excluded non-rural townships and rural townships with populations of less than 10,000. Of the remaining townships, 20 townships per county were randomly selected by a computer-generated random numbers method, resulting in a total of 80 townships. Finally, a list of all mothers with registered births within the past 6 months of the survey was obtained from the township health center in each sample township. A total of 842 postpartum women were identified and contacted by the research team, with the assistance of local township health center doctors or village doctors to invite. In total, 55 eligible postpartum women failed to enroll in the study due to out-migration or travel at the time of the survey, intellectual disability or mental illness that prevented ability to give informed consent, or refusal to participate. Of the 842 lactating postpartum women who enrolled in this study, 787 postpartum women completed all aspects of the questionnaire, with a response rate of 93.5%.”

Comment 7: Line 141-150: These sentences are unclear. Kindly revisit. 

Response: Thank you for pointing this out. We have updated the sentence in the Methods section of the revised manuscript page 10-11, lines 192-208 (revised text in italics): 

“The research team sampled rural mothers within the 0-6 months postpartum period living in one prefecture of Sichuan province, China, following a multi-stage cluster sampling protocol. First, four nationally-designated poverty counties were selected within the sample prefecture. Second, sample townships were chosen within each sample county. To select townships representative of typical rural areas, the sampling frame excluded non-rural townships and rural townships with populations of less than 10,000. Of the remaining townships, 20 townships per county were randomly selected by a computer-generated random numbers method, resulting in a total of 80 townships. Finally, a list of all mothers with registered births within the past 6 months of the survey was obtained from the township health center in each sample township. A total of 842 postpartum women were identified and contacted by the research team, with the assistance of local township health center doctors or village doctors to invite. In total, 55 eligible postpartum women failed to enroll in the study due to out-migration or travel at the time of the survey, intellectual disability or mental illness that prevented ability to give informed consent, or refusal to participate. Of the 842 lactating postpartum women who enrolled in this study, 787 postpartum women completed all aspects of the questionnaire, with a response rate of 93.5%.”

Comment 8: Line 181: I am not sure you can use the word “extensive” as your manuscript has not really demonstrated the “extensiveness” of your approach.

Response: Thank you for pointing this out. We apologize for the inadequate references provided leading to errors in wording, and we have added references to this section of the revised manuscript page 13, lines 258-260 (revised text in italics):

“After an extensive literature review, the research team developed a Breastfeeding Problems Questionnaire that encompasses the most prevalent problems associated with breastfeeding [23-31].”

REFERENCES: 

23. Demirci JR, Bogen DL. An Ecological Momentary Assessment of Primiparous Women’s Breastfeeding Behavior and Problems from Birth to 8 Weeks. J Hum Lact. 2017; 33(2): 285-295. https://doi.org/10.1177/0890334417695206 PMID: 28418803

24. Kronborg H, Væth M. How Are Effective Breastfeeding Technique and Pacifier Use Related to Breastfeeding Problems and Breastfeeding Duration? Birth. 2009; 36(1):34-42. https://doi.org/10.1111/j.1523-536X.2008.00293.x PMID: 19278381

25. Karaçam Z, Sağlık M. Breastfeeding problems and interventions performed on problems: Systematic review based on studies made in Turkey. Turk Pediatri Arsivi. 2018; 53(3). https://doi.org/10.5152/TurkPediatriArs.2018.6350 PMID: 30459512

26. Talbert AW, Ngari M, Tsofa B, Mramba L, Mumbo E, Berkley JA, et al. “When you give birth you will not be without your mother” A mixed methods study of advice on breastfeeding for first-time mothers in rural coastal Kenya. Int Breastfeed J. 2016; 11(1):1-9. https://doi.org/10.1186/s13006-016-0069-6 PMID: 27118984

27. Wagner EA, Chantry CJ, Dewey KG, Nommsen-Rivers LA. Breastfeeding concerns at 3 and 7 days postpartum and feeding status at 2 months. Pediatrics. 2013; 132(4):e865-e875. https://doi.org/10.1542/peds.2013-0724 PMID: 24062375

28. Odom EC, Li R, Scanlon KS, Perrine CG, Grummer-Strawn L. Reasons for earlier than desired cessation of breastfeeding. Pediatrics. 2013; 131(3):e726. https://doi.org/10.1542/peds.2012-1295 PMID: 23420922

29. Berridge K, McFadden K, Abayomi J, Topping J. Views of breastfeeding difficulties among drop-in-clinic attendees. Matern Child Nutr. 2005; 1(4):250-262. https://doi.org/10.1111/j.1740-8709.2005.00014.x PMID: 16881907

30. Sun K, Chen M, Yin Y, Wu L, Gao L. Why Chinese mothers stop breastfeeding: Mothers’ self-reported reasons for stopping during the first six months. J Child Heal Care. 2017;21:53–363. https://doi.org/10.1177/1367493517719160 PMID: 29119825

31. Liu P, Qiao L, Xu F, Zhang M, Wang Y, Binns CW. Factors associated with breastfeeding duration: A 30-month cohort study in Northwest China. J Hum Lact. 2013;35:583–591. https://doi.org/10.1177/0890334418812044 PMID: 23504474

Comment 9: Line 268; ‘November to December, 2019’ are you sure this was an enough time to conducted enumeration based on your study area coverage? Please see again and modify accordingly.

Response: Thank you for raising this point. From November to December 2019, our research team split into four teams to lead field surveys with enumerators in four sample counties (Langzhong, Nanbu, Yilong and Jialing) at the same time. As a result, we were able to complete the enumeration based on our study area coverage within a two-month period. 

Comment 10: Line 273; why we used only 20 participants as pilot study and what was your reference’s we used only 20 participants?

Response: Thank you for this comment. The pilot study for our survey was conducted in two non-sample townships within the sample counties. The purpose of the pilot study was to test the feasibility of the questionnaire and ensure that all questions were appropriate and understandable for rural mothers in our study area. Since this pilot study did not involve any data analysis, this study did not make any special requirements for the sample size. The sample size for this pilot study was based on a study by Li et al. (2019) which piloted their survey among 20 breastfeeding women. To clarify this point, we have added this reference and updated the language in the Methods section of the revised manuscript on page 17-18, lines 355-360 (revised text in italics):

“To ensure the accuracy and consistency of our data collection, a uniform training session was provided to enumerators; in addition, following Li et al. [15], a pilot study was conducted among twenty participants in two non-sample townships to ensure the survey was appropriate and understandable for rural mothers in the study area.” 

REFERENCES:

15. Li T, Guo N, Jiang H, Eldadah M. Breastfeeding Self-Efficacy Among Parturient Women in Shanghai: A Cross-Sectional Study. J Hum Lact. 2019 Aug 5;35(3):583–91. http://journals.sagepub.com/doi/10.1177/0890334418812044 PMID: 30517822

Comment 11: Age category; which age classification are used in the study, which was ’18-30 Vs >31’?

Response: Thank you for this question. The age categories in our study were based on those used in a 2014 study by Jemin Zhu et al., who explored factors influencing breastfeeding self-efficacy in urban China. Zhu’s study similarly divided Chinese mothers into two age groups using 30 years as a node. We have added this reference and a corresponding note in the Results section of the revised manuscript on page 21 in the notes to Table 1 (revised text in italics):

“We divided mothers into two age groups using 30 years as a node, following the methods of a previous study of BSE in urban China by Zhu et al. [41].”

REFERENCES:

41. Zhu J, Chan WCS, Zhou X, Ye B, He HG. Predictors of breast feeding self-efficacy among Chinese mothers: A cross-sectional questionnaire survey. Midwifery. 2014;30(6):705-11. https://doi.org/10.1016/j.midw.2013.12.008 PMID: 24439394

Comment 12: Line 305-306 and line 315-316; we used mean score ± standard deviation, sure that the data distribution is normal?

Response: Thank you for bringing this to our attention. The results of the Kolmogorov-Smirnov test for breastfeeding self-efficacy suggest that the data distribution is normal (Z= 1.082, P> 0.05). We have updated the sentence in the Results section of the revised manuscript on page 22, lines 413-417 (revised text in italics):

“The results of the Kolmogorov-Smirnov test suggest that the distribution of BSE scores is normal (Z=1.082, P> 0.05). The average BSE score among the participants is 55.95 (SD= 8.92), and the mean score for each item is 3.50 (SD= 0.56).”

The results of the Kolmogorov-Smirnov test for maternal age suggest that the data is not normally distributed (Z=2.314, P< 0.05) and that, instead, the median and interquartile range should be used to statistically describe maternal age. We have updated the relevant sentence in the Results section of the revised manuscript on page 20, lines 401-403 (revised text in italics):

“The results of the Kolmogorov-Smirnov for maternal age suggest that the distribution is not normal (Z=2.314, P< 0.05). The median age of the postpartum women was 27 years (IQR= 24~31).” 

In addition, we have added the results of the Kolmogorov-Smirnov test for other continuous variables including the Dennis BSE Framework variables in the Results section of the revised manuscript on page 28, lines 458-461 (revised text in italics): 

“Table 5 presents the descriptive statistics of the Dennis BSE Framework variables excluding breastfeeding problems. The results of the Kolmogorov-Smirnov test suggest that the distribution of all these variables is not normal (P< 0.05); we therefore use the median and interquartile range for our subsequent analysis.”

Comment 13: Discussion: your discussion could be further developed so it doesn’t appear like a repetition of your objectives.

Response: Thank you for your suggestion. We have enriched and developed the content of the Discussion section on page 33-36, lines 543-602 (revised text in italics):

“The empirical results of this study also found that women who had negative breastfeeding experiences had significantly lower BSE than those without such experiences. Three variables related to breastfeeding problems, including the child having trouble sucking or latching onto the breast, not producing enough milk, and milk taking too long to secrete, were all significantly associated with lower BSE. Such breastfeeding problems may also explain the diminished confidence in breastfeeding techniques reported among postpartum women in our study. Although few studies have examined the role of negative breastfeeding experiences in BSE, the findings align with the Dennis BSE framework, which theorizes that successful performance accomplishments increase BSE, whereas repeated failures or difficulties diminish it [8]. 

Early challenges with breastfeeding may be particularly salient for BSE among postpartum mothers in rural China. Previous research has shown that more than half of postpartum women in rural China experienced problems in the early stages of breastfeeding [45]. In our study, 28.5% of postpartum women experienced difficulty with latching during the first two weeks of breastfeeding, 47.1% experienced insufficient milk supply, and 37.2% experienced slow milk secretion. Postpartum women who encounter these problems in the early stages of breastfeeding may feel inadequate in their breastfeeding techniques and overwhelmed by challenges, thus reducing BSE. Moreover, although these problems can be alleviated by educating women on effective breastfeeding techniques, it is often difficult for postpartum women in rural areas to obtain relevant counseling and guidance [46]. When breastfeeding problems arise but cannot be solved in a timely and effective manner, postpartum women’s BSE decreases, and mothers may eventually give up breastfeeding [47]. Therefore, public health services in rural China should focus on helping new mothers resolve early problems they encounter during the breastfeeding process, especially insufficient milk, poor sucking or latching, and slow milk secretion. 

In contrast to breastfeeding problems, the results find that social support from significant others and family support for breastfeeding were both significantly associated with higher BSE among postpartum women in rural China. This finding is consistent with BSE studies internationally [48,49], as well as studies of self-efficacy in general, both of which find that social support can increase one’s coping abilities and competence [50]. This also aligns with the Dennis BSE framework, which suggests that verbal persuasion from family members, especially significant others, encourages mothers to continue breastfeeding their infants despite challenges [8]. As the closest and most important social network, family members are particularly important sources of emotional support for postpartum women in general [51] and in breastfeeding promotion specifically [52]. In addition to emotional support, postpartum women with higher levels of breastfeeding support receive relatively more practical assistance from family, which may help them to persist in breastfeeding [35]. In rural China, however, family members and significant others rarely receive education on breastfeeding or how to support breastfeeding mothers [41]. Educating family members about the importance of breastfeeding support for postpartum women may therefore increase BSE, motivation to breastfeed, and success in breastfeeding.”

REFERENCES:

8. Dennis C-L. Theoretical Underpinnings of Breastfeeding Confidence: A Self-Efficacy Framework. J Hum Lact. 1999 Sep;15(3):195–201. http://journals.sagepub.com/doi/10.1177/089033449901500303 PMID: 10578797

35. Zhu X, Liu L, Wang Y. Utilizing a Newly Designed Scale for Evaluating Family Support and Its Association with Exclusive Breastfeeding. Breastfeed Med. 2016 Dec;11(10):526–31. https://doi.org/10.1089/bfm.2016.0090 PMID: 27870578

41. Zhu J, Chan WCS, Zhou X, Ye B, He HG. Predictors of breast feeding self-efficacy among Chinese mothers: A cross-sectional questionnaire survey. Midwifery. 2014;30(6):705-11. https://doi.org/10.1016/j.midw.2013.12.008 PMID: 24439394

45. Zhang LF, Mu M, Nie W, Song SY, Gao QF, Nie JC. [Impact of infant formula sales promotion – recommendation and trial use on breastfeeding practice among mothers of 0 – 6 months infants in poverty-stricken rural areas of China]. Chin J Public Health. 2021; 37(02):280-5. Chinese.

46. Tang L, Binns CW, Luo C, Zhong Z, Lee AH. Determinants of breastfeeding at discharge in rural China. Asia Pac J Clin Nutr. 2013;22(3):443–8. https://doi.org/10.6133/apjcn.2013.22.3.20 PMID: 23945415

47. Nilsson IMS, Kronborg H, Rahbek K, Strandberg-Larsen K. The significance of early breastfeeding experiences on breastfeeding self-efficacy one week postpartum. Matern Child Nutr. 2020;16(3):1–12. https://doi.org/10.1111/mcn.12986 PMID: 32543045

48. Hinic K. Predictors of Breastfeeding Confidence in the Early Postpartum Period. JOGNN - J Obstet Gynecol Neonatal Nurs. 2016;45(5):649–60. http://dx.doi.org/10.1016/j.jogn.2016.04.010 PMID: 27472996

49. Mirghafourvand M, Malakouti J, Mohammad-Alizadeh-Charandabi S, Faridvand F. Predictors of Breastfeeding Self-efficacy in Iranian Women: A Cross-Sectional Study. Int J Womens Heal Reprod Sci. 2018;6(3):380–5. https://doi.org/10.15296/ijwhr.2018.62

50. Clapton-Caputo E, Sweet L, Muller A. A qualitative study of expectations and experiences of women using a social media support group when exclusively expressing breastmilk to feed their infant. Women and Birth. 2021;34(4),370–380. https://doi.org/10.1016/j.wombi.2020.06.010 PMID: 32674991

51. Bai DL, Fong DYT, Lok KYW, Tarrant M. Relationship between the Infant Feeding Preferences of Chinese Mothers’ Immediate Social Network and Early Breastfeeding Cessation. J Hum Lact. 2016;32:301–308. https://doi.org/10.1177/0890334416630537 PMID: 26887843

52. Kim JH, Fiese BH, Donovan SM. Breastfeeding is Natural but Not the Cultural Norm: A Mixed-Methods Study of First-Time Breastfeeding, African American Mothers Participating in WIC. J Nutr Educ Behav. 2017;49:S151. https://doi.org/10.1016/j.jneb.2017.04.003 PMID: 28689552

Comment 14: Line 464-466; this is the limitation of the study design itself not yours, please modify it. Why we are incorporating your strength?

Response: Thank you for pointing this out. We agree that this is the limitation of the study design itself not ours. We have removed this from the revised manuscript. 

Comment 15: General comment; There are several grammatical issues in manuscript which make it difficult for the readers to grasp the important points to convey in sections where they occur. I suggest you critically review the manuscript or possibly the service of copyeditor.

Response: Thank you for bringing this to our attention. We have accepted the service of an editor to critically review the manuscript for spelling, grammar and comprehension. All changes are highlighted yellow in the revised manuscript.

Response to the Journal Requirements:

Comment 1: *Please ensure that your manuscript meets PLOS ONE's style requirements, including those for file naming.

Response: We have critically reviewed the manuscript to ensure that the manuscript meets PLOS ONE’s style requirements, including those for file naming.

Comment 2: *We note that you have stated that you will provide repository information for your data at acceptance. Should your manuscript be accepted for publication, we will hold it until you provide the relevant accession numbers or DOIs necessary to access your data. If you wish to make changes to your Data Availability statement, please describe these changes in your cover letter and we will update your Data Availability statement to reflect the information you provide.

Response: Thank you for your note. We do not intend to change the data availability statement.

Comment 3: *Please review your reference list to ensure that it is complete and correct. If you have cited papers that have been retracted, please include the rationale for doing so in the manuscript text, or remove these references and replace them with relevant current references. Any changes to the reference list should be mentioned in the rebuttal letter that accompanies your revised manuscript. If you need to cite a retracted article, indicate the article’s retracted status in the References list and also include a citation and full reference for the retraction notice.

Response: We have reviewed our reference list to ensure that it is complete and correct and that no retracted papers have been cited. All changes are highlighted yellow in the revised manuscript.

We have removed the following references:

1. Semenic S, Loiselle C, Gottlieb L. Predictors of the duration of exclusive breastfeeding among first-time mothers. Res Nurs Heal. 2008; 31:428–441.https://doi.org/10.1002/nur.20275

2. Asgarian A, Hashemi M, Pournikoo M, Mirazimi TS, Zamanian H, Amini-Tehrani M. Translation, Validation, and Psychometric Properties of Breastfeeding Self-Efficacy Scale—Short Form Among Iranian Women. J Hum Lact. 2020;36(2):227–35. 

3. Petrozzi A, Gagliardi L. Breastfeeding self-efficacy scale: Validation of the Italian version and correlation with breast-feeding at 3 months. J Pediatr Gastroenterol Nutr. 2016;62(1):137–9.

4. Li S, Li L, Zheng H, Wang Y, Zhu X, Yang Y, et al. Relationship between multifaceted body image and negative affect among women undergoing mastectomy for breast cancer: a longitudinal study. Arch Womens Ment Health. 2018;21(6):681–8. doi:10.1007/s00737-018-0860-z

5. Jiang LC, Yan YJ, Jin ZS, Hu ML, Wang L, Song Y, et al. The Depression Anxiety Stress Scale-21 in Chinese Hospital Workers: Reliability, Latent Structure, and Measurement Invariance Across Genders. Front Psychol. 2020;11(March):1–9.

6. McCarter-Spaulding D, Gore R. Social Support Improves Breastfeeding Self-Efficacy in a Sample of Black Women. Clin Lact. 2012; https://doi.org/10.1891/215805312807022923

We have added the following references:

1. Glassman ME, McKearney K, Saslaw M, Sirota DR. Impact of breastfeeding self-efficacy and sociocultural factors on early breastfeeding in an urban, predominantly dominican community. Breastfeed Med. 2014;9(6):301–7. https://doi.org/10.1089/bfm.2014.0015 PMID: 24902047

2. Wu DS, Hu J, Mccoy TP, Efird JT. The effects of a breastfeeding self-efficacy intervention on short-term breastfeeding outcomes among primiparous mothers in Wuhan, China. J Adv Nurs. 2014;70(8):1867–79. https://doi.org/10.1111/jan.12349 PMID: 24400967

3. Liu Y, Li N, Mei Z, Li Z, Ye R, Zhang L, et al. Effects of prenatal micronutrients supplementation timing on pregnancy-induced hypertension: Secondary analysis of a double-blind randomized controlled trial. Matern Child Nutr. 2021;(January):e13157. https://doi.org/10.1111/mcn.13157 PMID: 33594802

4. Demirci JR, Bogen DL. An Ecological Momentary Assessment of Primiparous Women’s Breastfeeding Behavior and Problems from Birth to 8 Weeks. J Hum Lact. 2017; 33(2): 285-295. https://doi.org/10.1177/0890334417695206 PMID: 28418803

5. Kronborg H, Væth M. How Are Effective Breastfeeding Technique and Pacifier Use Related to Breastfeeding Problems and Breastfeeding Duration? Birth. 2009; 36(1):34-42. https://doi.org/10.1111/j.1523-536X.2008.00293.x PMID: 19278381

6. Karaçam Z, Sağlık M. Breastfeeding problems and interventions performed on problems: Systematic review based on studies made in Turkey. Turk Pediatri Arsivi. 2018; 53(3). https://doi.org/10.5152/TurkPediatriArs.2018.6350 PMID: 30459512

7. Talbert AW, Ngari M, Tsofa B, Mramba L, Mumbo E, Berkley JA, et al. “When you give birth you will not be without your mother” A mixed methods study of advice on breastfeeding for first-time mothers in rural coastal Kenya. Int Breastfeed J. 2016; 11(1):1-9. https://doi.org/10.1186/s13006-016-0069-6 PMID: 27118984

8. Wagner EA, Chantry CJ, Dewey KG, Nommsen-Rivers LA. Breastfeeding concerns at 3 and 7 days postpartum and feeding status at 2 months. Pediatrics. 2013; 132(4):e865-e875. https://doi.org/10.1542/peds.2013-0724 PMID: 24062375

9. Odom EC, Li R, Scanlon KS, Perrine CG, Grummer-Strawn L. Reasons for earlier than desired cessation of breastfeeding. Pediatrics. 2013; 131(3):e726. https://doi.org/10.1542/peds.2012-1295 PMID: 23420922

10. Berridge K, McFadden K, Abayomi J, Topping J. Views of breastfeeding difficulties among drop-in-clinic attendees. Matern Child Nutr. 2005; 1(4):250-262. https://doi.org/10.1111/j.1740-8709.2005.00014.x PMID: 1688190710. 

11. Zhang LF, Mu M, Nie W, Song SY, Gao QF, Nie JC. [Impact of infant formula sales promotion – recommendation and trial use on breastfeeding practice among mothers of 0 – 6 months infants in poverty-stricken rural areas of China]. Chin J Public Health. 2021; 37(02):280-5. Chinese.

12. Tang L, Binns CW, Luo C, Zhong Z, Lee AH. Determinants of breastfeeding at discharge in rural China. Asia Pac J Clin Nutr. 2013;22(3):443–8. https://doi.org/10.6133/apjcn.2013.22.3.20 PMID: 23945415

13. Clapton-Caputo E, Sweet L, Muller A. A qualitative study of expectations and experiences of women using a social media support group when exclusively expressing breastmilk to feed their infant. Women and Birth. 2021;34(4),370–380. https://doi.org/10.1016/j.wombi.2020.06.010 PMID: 32674991

---

## [Decision Letter · Decision Letter 1]

18 Mar 2022

Determinants of breastfeeding self-efficacy among postpartum women in rural China: A cross-sectional study

PONE-D-21-31115R1

Dear Dr. Zhou,

We’re pleased to inform you that your manuscript has been judged scientifically suitable for publication and will be formally accepted for publication once it meets all outstanding technical requirements.

Kind regards,

Wubet Alebachew Bayih, M.Sc.

Academic Editor

PLOS ONE

Additional Editor Comments (optional):

Reviewers' comments:

Reviewer's Responses to Questions

**Comments to the Author**

1. If the authors have adequately addressed your comments raised in a previous round of review and you feel that this manuscript is now acceptable for publication, you may indicate that here to bypass the “Comments to the Author” section, enter your conflict of interest statement in the “Confidential to Editor” section, and submit your "Accept" recommendation.

Reviewer #1: All comments have been addressed

Reviewer #2: All comments have been addressed

2. Is the manuscript technically sound, and do the data support the conclusions?

Reviewer #1: Yes

Reviewer #2: Partly

3. Has the statistical analysis been performed appropriately and rigorously? 

Reviewer #1: Yes

Reviewer #2: Yes

4. Have the authors made all data underlying the findings in their manuscript fully available?

Reviewer #1: Yes

Reviewer #2: No

5. Is the manuscript presented in an intelligible fashion and written in standard English?

Reviewer #1: Yes

Reviewer #2: Yes

6. Review Comments to the Author

Reviewer #1: Authors have made a substantial improvement in their revised document. It now looks modified and clear to the reader.

Reviewer #2: 1.General comment; There are several grammatical issues in manuscript which make it difficult for the readers to grasp the important points to convey in sections where they occur. I suggest you critically review the manuscript or possibly the service of copyeditor.

7. PLOS authors have the option to publish the peer review history of their article (what does this mean?). If published, this will include your full peer review and any attached files.

Reviewer #1: **Yes: **Hassen Mosa Halil(MSc)

Reviewer #2: **Yes: **Bekalu Getnet Kassa

---

## [Editor Report · Acceptance letter]

29 Mar 2022

PONE-D-21-31115R1 

Determinants of breastfeeding self-efficacy among postpartum women in rural China: A cross-sectional study 

Dear Dr. Zhou:

I'm pleased to inform you that your manuscript has been deemed suitable for publication in PLOS ONE. Congratulations! Your manuscript is now with our production department. 

Kind regards, 

on behalf of

Dr. Wubet Alebachew Bayih 

Academic Editor

PLOS ONE